# Learning Affinity via Spatial Propagation Networks

**Sifei Liu**
UC Merced, NVIDIA

**Shalini De Mello**
NVIDIA

**Jinwei Gu**
NVIDIA

**Guangyu Zhong**
Dalian University of Technology

**Ming-Hsuan Yang**
UC Merced, NVIDIA

**Jan Kautz**
NVIDIA

## Abstract

In this paper, we propose spatial propagation networks for learning the affinity matrix for vision tasks. We show that by constructing a row/column linear propagation model, the spatially varying transformation matrix exactly constitutes an affinity matrix that models dense, global pairwise relationships of an image. Specifically, we develop a three-way connection for the linear propagation model, which (a) formulates a sparse transformation matrix, where all elements can be outputs from a deep CNN, but (b) results in a dense affinity matrix that effectively models any task-specific pairwise similarity matrix. Instead of designing the similarity kernels according to image features of two points, we can directly output all the similarities in a purely data-driven manner. The spatial propagation network is a generic framework that can be applied to many affinity-related tasks, such as image matting, segmentation and colorization, to name a few. Essentially, the model can learn semantically-aware affinity values for high-level vision tasks due to the powerful learning capability of deep CNNs. We validate the framework on the task of refinement of image segmentation boundaries. Experiments on the HELEN face parsing and PASCAL VOC-2012 semantic segmentation tasks show that the spatial propagation network provides a general, effective and efficient solution for generating high-quality segmentation results.

## 1 Introduction

An affinity matrix is a generic matrix that determines how close, or similar, two points are in a space. In computer vision tasks, it is a weighted graph that regards each pixel as a node, and connects each pair of pixels by an edge [25, 16, 15, 10, 29]. The weight on that edge should reflect the pairwise similarity with respect to different tasks. For example, for low-level vision tasks such as image filtering, the affinity values should reveal the low-level coherence of color and texture [29, 28, 10, 9]; for mid to high-level vision tasks such as image matting and segmentation [16, 22], the affinity measure should reveal the semantic-level pairwise similarities. Most techniques explicitly or implicitly assume a measurement or a similarity structure over the space of configurations. The success of such algorithms depends heavily on the assumptions made to construct these affinity matrices, which are generally not treated as part of the learning problem.

In this paper, we show that the problem of learning the affinity matrix can be equivalently expressed as learning a group of small row/column-wise, spatially varying linear transformation matrices. Since a linear transformation can be easily implemented as a differentiable module in a deep neural network, the transformation matrix can be learned in a purely data-driven manner as opposed to being constructed by hand. Specifically, we adopt an independent deep CNN with the original RGB images as inputs to output all entities of the matrix, such that the affinity is learned by a deep model conditioned on the specific inputs. We show that using a three-way connection, instead of the full connection between adjoining rows/columns, is sufficient for learning a dense affinity matrix and requires much fewer output channels of a deep CNN. Therefore, instead of using designed features and kernel tricks, our network outputs all entities of the affinity matrix in a data-driven manner.

The advantages of learning an affinity matrix in a data-driven manner are multifold. First, a hand-designed similarity matrix based on a distance metric in a certain space (*e.g.*, RGB or Euclidean [10, 25, 5, 36, 14]) may not adequately describe the pairwise relationships in the mid-to-high-level feature spaces. To apply such designed pairwise kernels to tasks such as semantic segmentation, multiple iterations are required [14, 5, 36] for satisfactory performance. In contrast, the proposed method learns and outputs all entities of an affinity matrix under direct supervision of ultimate objectives, where no iteration, specific design or assumption about the kernel function is needed. Second, we can learn the high-level semantic affinity measures by initializing with hierarchical deep features from pre-trained VGG [26] and ResNet [11] networks where conventional metrics and kernels may not be applied. Due to the above properties, the framework is far more efficient than the related graphical models, such as Dense CRF.

Our proposed architecture, namely spatial propagation network (SPN), contains a deep CNN that learns the entities of the affinity matrix and a spatial linear propagation module, which propagates information in an image using the learned affinity values. Images or general 2D matrices are input into the module, and propagated under the guidance of the learned affinity values. All modules are differentiable and jointly trained using the stochastic gradient descent (SGD) method. The spatial linear propagation module is computationally efficient for inference due to the linear time complexity of its recurrent architecture.

## 2   Related Work

Numerous methods explicitly design affinity matrices for image filtering [29, 10], colorization [15], matting [16] and image segmentation [14] based on the characterstics of the problem. Other methods, such as total variation (TV) [23] and learning to diffuse [18] improve the modeling of pairwise relationships by utilizing different objectives, or incorporating more priors into diffusion partial differential equations (PDEs). However, due to the lack of an effective learning strategy, it is still challenging to produce learning-based affinity for complex visual analysis problems. Recently, Maire et al. [22] trained a deep CNN to directly predict the entities of an affinity matrix, which demonstrated good performance on image segmentation. However, since the affinity is followed by a solver of spectral embedding as an independent part, it is not directly supervised for the classification/prediction task. Bertasius et al. [2] introduced a random walk network that optimizes the objectives of pixel-wise affinity for semantic segmentation. Differently, their affinity matrix is additionally supervised by ground-truth sparse pixel similarities, which limits the potential connections between pixels.

On the other hand, many graphical model-based methods have successfully improved the performance of image segmentation. In the deep learning framework, conditional random fields (CRFs) with efficient mean field inference are frequently used [14, 36, 17, 5, 24, 1] to model the pairwise relations in the semantic labeling space. Some methods use CFR as a post-processing module [5], while others integrate it as a jointly-trained part [36, 17, 24, 1]. While both methods describe the densely connected pairwise relationships, dense CRFs rely on designed kernels, while our method directly learns all pairwise links. Since in this paper, SPN is trained as a universal segmentation refinement module, we specifically compare it with one of the methods [5] that relies on dense CRF [14] as a post-processing strategy. Our architecture is also related to the multi-dimensional RNN or LSTM [30, 3, 8]. However, both the standard RNN and LSTM contain multiple non-linear units and thus do not fit into our proposed affinity framework.

## 3   Proposed Approach

In this work, we construct a spatial propagation network that can transform a two-dimensional (2D) map (*e.g.*, coarse image segmentation) into a new one with desired properties (*e.g.*, refined segmentation). With spatially varying parameters that supports the propagation process, we show theoretically in Section 3.1 that this module is equivalent to the standard anisotropic diffusion process [32, 18]. We prove that the transformation of maps is controlled by a Laplacian matrix that is constituted by the parameters of the spatial propagation module. Since the propagation module is differentiable, its parameters can be learned by any type of neural network (*e.g.*, a typical deep CNN) that is connected to this module, through joint training. We introduce the spatial propagation network in Section 3.2, and specifically analyze the properties of different types of connections within its framework for learning the affinity matrix.

### 3.1 Linear Propagation as Spatial Diffusion

We apply a linear transformation by means of the spatial propagation network, where a matrix is scanned row/column-wise in four fixed directions: left-to-right, top-to-bottom, and verse-vise. This strategy is used widely in [8, 30, 19, 4]. We take the left-to-right direction as an example for the following discussion. Other directions are processed independently in the same manner.

We denote $X$ and $H$ as two 2D maps of size $n \times n$, with exactly the same dimensions as the matrix before and after spatial propagation, where $x_t$ and $h_t$, respectively, represent their $t^{th}$ columns with $n \times 1$ elements each. We linearly propagate information from left-to-right between adjacent columns using an $n \times n$ linear transformation matrix $w_t$ as:

$$h_t = (I - d_t)\, x_t + w_t h_{t-1}, \quad t \in [2, n] \tag{1}$$

where $I$ is the $n \times n$ identity matrix, the initial condition $h_1 = x_1$, and $d_t(i, i)$ is a diagonal matrix, whose $i^{th}$ element is the sum of all the elements of the $i^{th}$ row of $w_t$ except $w_t(i, j)$ as:

$$d_t(i, i) = \sum_{j=1, j \neq i}^{n} w_t(i, j). \tag{2}$$

To propagate across the entire image, the matrix $H$, where $\{h_t \in H, t \in [1, n]\}$, is updated in a column-wise manner recursively. For each column, $h_t$ is a linear, weighted combination of the previous column $h_{t-1}$, and the corresponding column $x_t$ in $X$. When the recursive scanning is finished, the updated 2D matrix $H$ can be expressed with an expanded formulation of Eq. (1):

$$H_v = \begin{bmatrix} I & 0 & \cdots & \cdots & 0 \\ w_2 & \lambda_2 & 0 & \cdots & \cdots \\ w_3 w_2 & w_3 \lambda_2 & \lambda_3 & 0 & \cdots \\ \vdots & \vdots & \vdots & \ddots & \vdots \\ \vdots & \vdots & \cdots & \cdots & \lambda_n \end{bmatrix} X_v = G X_v, \tag{3}$$

where $G$ is a lower triangular, $N \times N (N = n^2)$ transformation matrix, which relates $X$ and $H$. $H_v$ and $X_v$ are vectorized versions of $X$ and $H$, respectively, with the dimension of $N \times 1$. Specifically, they are created by concatenating $h_t$ and $x_t$ along the same, single dimension, *i.e.*, $H_v = \begin{bmatrix} h_1^T, ..., h_n^T \end{bmatrix}^T$ and $X_v = \begin{bmatrix} x_1^T, ..., x_n^T \end{bmatrix}^T$. All the parameters $\{\lambda_t, w_t, d_t, I\}, t \in [2, n]$ are $n \times n$ sub-matrices, where $\lambda_t = I - d_t$.

In the following section, we validate that Eq. (3) can be expressed as a spatial anisotropic diffusion process, with the corresponding propagation affinity matrix constituted by all $w_t$ for $t \in [2, n]$.

**Theorem 1.** *The summation of elements in each row of $G$ equals to one.*

Since G contains $n \times n$ sub-matrices, each representing the transformation between the corresponding columns of $H$ and $X$, we denote all the weights used to compute $h_t$ as the $t^{th}$ block-row $G_t$. On setting $\lambda_1 = I$, the $k^{th}$ constituent $n \times n$ sub-matrix of $G_t$ is:

$$G_{tk} = \begin{cases} \displaystyle\prod_{\tau=k+1}^{t} w_\tau \lambda_k, & k \in [1, t-1] \\ \lambda_k, & k = t \end{cases} \tag{4}$$

To prove that the summation of any row in $G$ equals to one, we instead prove that for $\forall t \in [1, n]$, each row of $G_t$ has the summation of one.

*Proof.* Denoting $E = [1, 1, ..., 1]^T$ as an $n \times 1$ vector, we need to prove that $G_t [1, ..., 1]_{N \times 1}^T = E$. Equivalently $\sum_{k=1}^{t} G_{tk} E = E$, because $G$ is a lower triangular matrix. In the following part, we first prove that when $m \in [1, t-1]$, we have $\sum_{k=1}^{m} G_{tk} E = \prod_{\tau=m+1}^{t} w_t E$ by mathematical induction .

**Initial step.** When $m = 1$, $\sum_{k=1}^{m} G_{tk} E = G_{t1} E = \prod_{\tau=2}^{t} w_\tau E$, which satisfies the assertion.

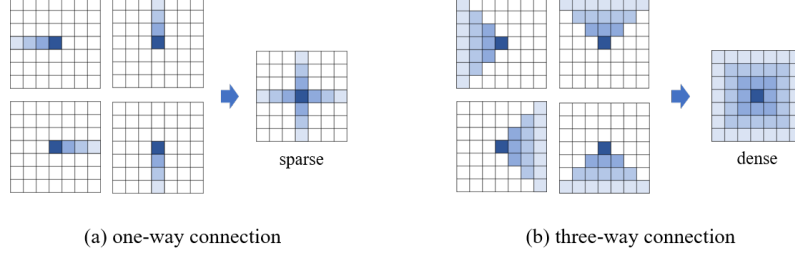

| (a) one-way connection | (b) three-way connection |

Figure 1: Different propagation ranges for (a) one-way connections; and (b) three-way connections. Each pixel (node) receives information from a single line with one-way connection, and from a 2 dimensional plane with three-way connection. Integration of four directions w.r.t. (a) results in global, but sparsely connected pairwise relations, while (b) formulates global and densely connected pairwise relations.

**Inductive step.** Assume there is an $n \in [1, t-1]$, such that $\sum_{k=1}^{n} G_{tk} E = \prod_{\tau=n+1}^{t} w_t E$, we must prove the formula is true for $n + 1 \in [1, t-1]$.

$$\sum_{k=1}^{n+1} G_{tk} E = \sum_{k=1}^{n} G_{tk} E + G_{t(n+1)} E = \prod_{\tau=n+1}^{t} w_\tau E + \prod_{\tau=n+2}^{t} w_\tau = \prod_{\tau=n+2}^{t} w_\tau \left[ (w_{n+1} + I - d_{n+1}) E \right].$$
(5)

According to the formulation of the diagonal matrix in Eq. (2) we have $\sum_{k=1}^{n+1} G_{tk} E = \prod_{\tau=n+2}^{t} w_\tau E$. Therefore, the assertion is satisfied. When $m = t$, we have:

$$\sum_{k=1}^{t} G_{tk} E = \sum_{k=1}^{t-1} G_{tk} E + G_{tt} E = \prod_{\tau=t}^{t} w_\tau E + \lambda_t E = w_\tau E + (I - d_t) E = E,$$
(6)

which yields the equivalence of Theorem 1. $\square$

**Theorem 2.** *We define the evolution of a 2D matrix as a time sequence $\{U\}_T$, where $U(T = 1) = U_1$ is the initial state. When the transformation between any two adjacent states follows Eq. (3), the sequence is a diffusion process expressed with a partial differential equation (PDE):*

$$\partial_T U = -LU$$
(7)

*where $L = D - A$ is the Laplacian matrix, $D$ is the degree matrix composed of $d_t$ in Eq. (2), and $A$ is the affinity matrix composed by the off-diagonal elements of $G$.*

*Proof.* We substitute the $X$ and $H$ as two consecutive matrices $U_{T+1}$ and $U_T$ in (3). According to Theorem 1, we ensure that the sum of each row $I - G$ is 0 that can formulate a standard Laplacian matrix. Since $G$ has the diagonal sub-matrix $I - d_t$, we can rewrite (3) as:

$$U_{T+1} = (I - D + A) U_T = (I - L) U_T$$
(8)

where $G = (I - D + A)$, $D$ is an $N \times N$ diagonal matrix containing all the $d_t$ and $A$ is the off-diagonal part of $G$. It then yields $U_{T+1} - U_T = -LU_T$, a discrete formulation of (7) with the time discretization interval as one. $\square$

Theorem 2 shows the essential property of the row/column-wise linear propagation in Eq. (1): it is a standard diffusion process where $L$ defines the spatial propagation and $A$, the affinity matrix, describes the similarities between any two points. Therefore, learning the image affinity matrix $A$ in Eq. (8) is **equivalent** to learning a group of transformation matrices $w_t$ in Eq. (1).

In the following section, we show how to build the spatial propagation (1) as a differentiable module that can be inserted into a standard feed-forward neural network, so that the affinity matrix $A$ can be learned in a data-driven manner.

## 3.2 Learning Data-Driven Affinity

Since the spatial propagation in Eq.(1) is differentiable, the transformation matrix can be easily configured as a row/column-wise fully-connected layer. However, we note that since the affinity matrix indicates the pairwise similarities of a specific input, it should also be conditioned on the

content of this input (*i.e.*, different input images should have different affinity matrices). Instead of setting the $w_t$ matrices as fixed parameters of the module, we design them as the outputs of a deep CNN, which can be directly conditioned on an input image.

One simple way is to set the output of the deep CNN to use the same size as the input matrix. When the input has $c$ channels (*e.g.*, an RGB image has $c = 3$), the output needs $n \times c \times 4$ channels (there are $n$ connections from the previous row/column per pixel per channel, and with four different directions). Obviously, this is too many (*e.g.*, an $128 \times 128 \times 16$ feature map needs an output of $128 \times 128 \times 8192$) to be implemented in a real-world system. Instead of using full connections between the adjacent rows/columns, we show that certain local connections, corresponding to a sparse row/column-wise transform matrix, can also formulate densely connected affinity. Specifically, we introduce the (a) one-way connection and the (b) three-way connection as two different ways to implement Eq. (1).

**One-way connection.** The one-way connection enables every pixel to connect to only one pixel from the previous row/column (see Figure 1(a)). It is equivalent to one-dimensional (1D) linear recurrent propagation that scans each row/column independently as a 1D sequence. Following Eq. (1), we denote $x_{k,t}$ and $h_{k,t}$ as the $k^{th}$ pixels in the $t^{th}$ column, where the left-to-right propagation for one-way connection is:

$$h_{k,t} = (1 - p_{k,t}) \cdot x_{k,t} + p_{k,t} \cdot h_{k,t-1}, \tag{9}$$

where $p$ is a scaler weight indicating the propagation strength between the pixels at $\{k, t-1\}$ and $\{k, t\}$. Equivalently, $w_t$ in Eq. (1) is a diagonal matrix, with the elements constituted by $p_{k,t}, k \in [1, n]$. The one-way connection is a direct extension of sequential recurrent propagation [8, 31, 13]. The exact formulation of Eq. (9) has been used previously for semantic segmentation [4] and for learning low-level vision filters [19]. In [4], Chen *et al.*explain it as domain transform, where for semantic segmentation, $p$ corresponds to the object edges. Liu *et al.* [19] explain it by arbitrary-order recursive filters, where $p$ corresponds to more general image properties (*e.g.*, low-level image/color edges, missing pixels, etc.). Both of these formulations can be explained as the same linear propagation framework of Eq. (1) with one-way connections.

**Three-way connection.** We propose a novel three-way connection in this paper. It enables each pixel to connect to three pixels from the previous row/column, *i.e.*, the left-top, middle and bottom pixels from the previous column for the left-to-right propagation direction (see Figure. 2(b)). With the same notations, we denote $\mathbb{N}$ as the set of these three pixels. Then the propagation for the three-way connection is:

$$h_{k,t} = \left(1 - \sum_{k \in \mathbb{N}} p_{k,t}\right) x_{k,t} + \sum_{k \in \mathbb{N}} p_{k,t} h_{k,t-1} \tag{10}$$

Equivalently, $w_t$ forms a tridiagonal matrix, with $p_{:,k}, k \in \mathbb{N}$ constitute the three non-zero elements of each row/column.

**Relations to the affinity matrix.** As introduced in Theorem 2, the affinity matrix $A$ with linear propagation is composed of the off-diagonal elements of $G$ in Eq. (3). The one-way connection formulates a spares affinity matrix, since each sub-matrix of $A$ has nonzero elements only along its diagonal, and the multiplication of several individual diagonal matrics will also results in a diagonal matrix. On the other hand, the three-way connection, also with a sparse $w_t$, can form a relatively dense $A$ with the multiplication of several different tridiagonal matrices. It means pixels can be densely and globally associated, by simply increasing the number of connections of each pixel during spatial propagation from one to three. As shown in Figures 2(a) and 2(b), the propagation of one-way connections is restricted to a single row, while the three-way connections can expand the region to a triangular 2D plane with respect to each direction. The summarization of the four directions result in dense connections of all pixels to each other (see Figure. 2(b)).

**Stability of linear propagation.** Model stability is of critical importance for designing linear systems. In the context of spatial propagation (Eq. 1), it refers to restricting the responses or errors that flow in the module from going to infinity, and preventing the network from encountering the vanishing of gradients in the backpropagation process [37]. Specifically, the norm of the temporal Jacobian $\partial h_t \setminus \partial h_{t-1}$ should be equal to or less than one. In our case, it is equivalent to regularizing each transformation matrix $w_t$ with its norm satisfying

$$\|\partial h_t \setminus \partial h_{t-1}\| = \|w_t\| \leq \lambda_{max}, \tag{11}$$

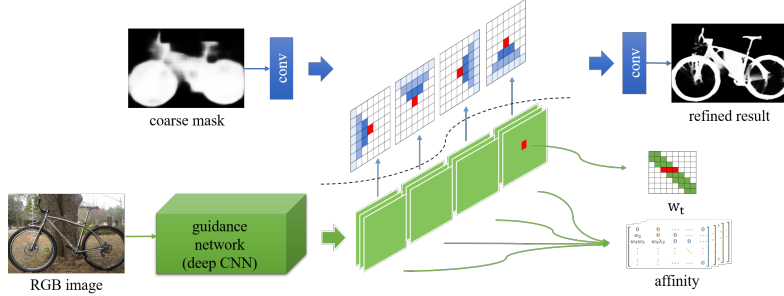

coarse mask

refined result

$w_t$

RGB image

affinity

Figure 2: We illustrate the general architecture of the SPN using a three-way connection for segmentation refinement. The network, divided by the black dash line, contains a propagation module (upper) and a guidance network (lower). The guidance network outputs all entities that can constitute four affinity matrices, where each sub-matrix $w_t$ is a tridiagonal matrix. The propagation module, being guided by the affinity matrices, deforms the input mask to a desired shape. All modules are differentiable and jointly learned via SGD.

where $\lambda_{max}$ denotes the largest singularity value of $w_t$. This condition, $\lambda_{max} \leq 1$ provides a sufficient condition for stability.

**Theorem 3.** *Let $\left\{p_{t,k}^K\right\}_{k\in\mathbb{N}}$ be the weight in $w_t$, the model can be stabilized if $\sum_{k\in\mathbb{N}}\left|p_{t,k}^K\right| \leq 1$. See the supplementary material for proof.*

Theorem 3 shows that the stability of a linear propagation model can be maintained by regularizing the all weights of each pixel in the hidden layer $H$, with the summation of their absolute values less than one. For the one-way connection, Chen *et al.* [4] limited each scalar output $p$ to be within $(0,1)$. Liu *et al.* [19] extended the range to $(-1,1)$, where the negative weights showed preferable effects for learning image enhancers. It indicates that the affinity matrix is not necessarily restricted to be positive/semi-positive definite (*e.g.*, the setting is also applied in [16].) For the three-way connection, we simply regularize the three weights (the output of a deep CNN) according to Theorem 3 without restriction to be any positive/semi-positive definite.

## 4   Implementation

We specify two separate branches: (a) a deep CNN, namely the guidance network that outputs all elements of the transformation matrix, and (b) a linear propagation module that outputs the transformation matrix entities (see Figure 2). The propagation module receives an input map and output a refined or transformed result. It also takes the weights learned by the deep CNN guidance network as the second input. The structure of a guidance network can be any regular CNN, which is designed for the task at hand. Examples of this network are described in Section 5. It takes, as input, any 2D matrix that can help with learning the affinity matrix (*e.g.*, typically an RGB image), and outputs all the weights that constitute the transformation matrix $w_t$.

Suppose that we have a map of size $n \times n \times c$ that is input into the propagation module, the guidance network needs to output a weight map with the dimensions of $n \times n \times c \times (3 \times 4)$, *i.e.*, each pixel in the input map is paired with 3 scalar weights per direction, and 4 directions in total. The propagation module contains 4 independent hidden layers for the different directions, where each layer combines the input map with its respective weight map using Eq. (10). All submodules are differentiable and jointly trained using stochastic gradient descent (SGD). We use node-wise max-pooling [19] to integrate the hidden layers and to obtain the final propagation result.

We implement the network with a modified version of CAFFE [12]. We employ a parallel version of the SPN implemented in CUDA for propagating each row/column to the next one. We use the SGD optimizer, and set the base learning rate to 0.0001. In general, we train the networks for the HELEN and VOC segmentation tasks for about 40 and 100 epochs, respectively. The inference time (we do not use cuDNN) of SPN on HELEN and Pascal VOC is about 7ms and 84ms for an image of size $512 \times 512$ pixels, respectively. In comparison, the dense CRF (CPU only) takes about 1s [14], 3.2s [5] and 4.4s [36] with different publicly available implementations. We note that the majority of the time for the SPN is spend in the guidance network, which can be accelerated by utilizing various existing network compressing strategies, applying smaller models, or sharing weights with the segmentation model if they are trained jointly. During inference, a single $64 \times 64 \times 32$ SPN hidden layer takes 1.3ms with the same computational settings.

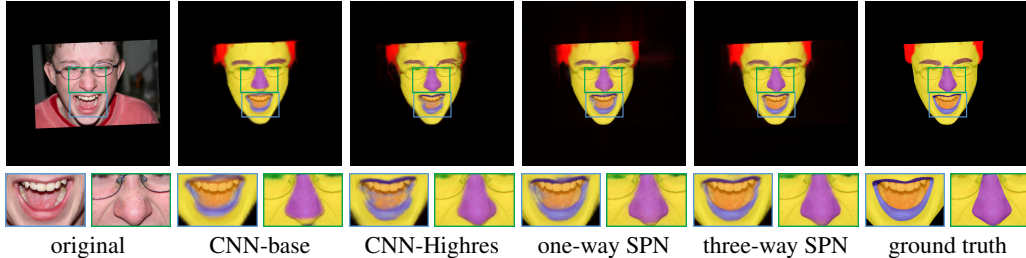

original     CNN-base     CNN-Highres     one-way SPN     three-way SPN     ground truth

Figure 3: Results of face parsing on the HELEN dataset with detailed regions cropped from the high resolution images. (The images are all in high resolution and can be viewed by zooming in.)

## 5 Experimental Results

The SPN can be trained jointly with any segmentation CNN model by being inserted on top of the last layer that outputs probability maps, or trained separately as a segmentation refinement model. In this paper we choose the second option. Given a coarse image segmentation mask as the input to the spatial propagation module, we show that the SPN can produce higher-quality masks with significantly refined details at object boundaries. Many models [21, 5] generate low-resolution segmentation masks with coarse boundary shapes to seek a balance between computational efficiency and semantic accuracy. The majority of work [21, 5, 36] choose to first produce an output probability map with $8\times$ smaller resolution, and then refine the result using either post-processing [5] or jointly trained modules [36]. Hence, producing high-quality segmentation results with low computational complexity is a non-trivial task. In this work, we train only one SPN model for a specific task, and treat it as a universal refinement tool for the different publicly available CNN models for each of these tasks.

We carry out the refinement of segmentation masks on two tasks: (a) generating high-resolution segmentations on the HELEN face parsing dataset [27]; and (b) refining generic object segmentation maps generated by pretrained models (*e.g.*, VGG based model [21, 5]. For the HELEN dataset, we directly use low-resolution RGB face images to train a baseline parser, which successfully encapsulates the global semantic information. The SPN is then trained on top of the coarse segmentations to generate high-resolution outputs. For the Pascal VOC dataset, we train the SPN on top of the coarse segmentation results generated by the FCN-8s [21], and directly generalize it to any other pretrained model.

**General network settings.** For both tasks, we train the SPN as a patch refinement model on top of the coarse map with basic semantic information. It is trained with smaller patches cropped from the original high-resolution images, their corresponding coarse segmentation maps produced by a baseline segmentor, and with the corresponding high-resolution ground-truth segmentation masks for supervision. All coarse segmentation maps are obtained by applying a baseline (for HELEN) or pre-trained (for Pascal VOC) image segmentation CNN to their standard training splits [6, 5]. Since the baseline HELEN parser produces low-resolution segmentation results, we upsample them using a bi-linear filter to be of the same size as the desired higher output resolution. We fix the size of our input patches to $128 \times 128$, use the *softmax* loss, and use the SGD solver for all the experiments. During training, the patches are sampled from image regions that contain more than one ground-truth segmentation label (*e.g.*, a patch with all pixels labeled as "background" will not be sampled). During testing, for the VOC dataset, we restrict the classes in the refined results to be contained within the corresponding coarse input. More specific settings are specified in the supplementary material.

**HELEN Dataset.** The HELEN dataset provides high-resolution photography-style face images (2330 in total), with high-quality manually labeled facial components including eyes, eyebrows, nose, lips, and jawline, which makes the high-resolution segmentation tasks applicable. All previous work utilize low-resolution parsing output as their final results for evaluation. Although many [27, 33, 20] achieve preferable performance, their results cannot be directly adopted by high-quality facial image editing applications. We use the same settings as the state-of-the work [20]. We use similarity transformation according to the results of 5-keypoint detection [35] to align all face images to the center. Keeping the original resolution, we then crop or pad them to the size of $1024 \times 1024$.

Table 1: Quantitative evaluation results on the HELEN dataset. We denote the upper and lower lips as "U-lip" and "L-lip", and overall mouth part as "mouth", respectively. The label definitions follow [20].

| Method | skin | brows | eyes | nose | mouth | U-lip | L-lip | in-mouth | overall |
|---|---|---|---|---|---|---|---|---|---|
| Liu *et al.* [20] | 90.87 | 69.89 | 74.74 | 90.23 | 82.07 | 59.22 | 66.30 | 81.70 | 83.68 |
| baseline-CNN | 90.53 | 70.09 | 74.86 | 89.16 | 83.83 | 55.61 | 64.88 | 71.72 | 82.89 |
| Highres-CNN | 91.78 | 71.84 | 74.46 | 89.42 | 81.83 | 68.15 | 72.00 | 71.95 | 83.21 |
| SPN (one-way) | 92.26 | 75.05 | 85.44 | 91.51 | 88.13 | 77.61 | 70.81 | 79.95 | 87.09 |
| SPN (three-way) | **93.10** | **78.53** | **87.71** | **92.62** | **91.08** | **80.17** | **71.63** | **83.13** | **89.30** |

We first train a baseline CNN with a symmetric U-net structure, where both the input image and the output map are $8\times$ smaller than the original image. The detailed settings are in the supplementary meterial. We apply the multi-objective loss as [20] to improve the accuracy along the boundaries. We note that the symmetric structure is powerful, since the results we obtained for the baseline CNN are comparable (see Table. 1) to that of [20], who apply a much larger model (38 MB vs. 12 MB). We then train a SPN on top of the baseline CNN results on the training set, with patches sampled from the high-resolution input image and the coarse segmentations masks. For the guidance network, we use the same structure as that of the baseline segmentation network, except that its upsampling part ends at a resolution of $64 \times 64$, and its output layer has $32 \times 12 = 384$ channels. In addition, we train another face parsing CNN with $1024 \times 1024$ sized inputs and outputs (CNN-Highres) for better comparison. It has three more sub-modules at each end of the baseline network, where all are configured with 16 channels to process higher resolution images.

We show quantitative and qualitative results in Table. 1 and 3 respectively. We compared the one/three way connection SPNs with the baseline, the CNN-Highres and the most relevant state-of-the-art technique for face parsing [20]. Note that the results of baseline and [20][1] are bi-linearly upsampled to $1024 \times 1024$ before evaluation. Overall, both SPNs outperform the other techniques with a significant margin of over 6 intersection-over-union (IoU) points. Especially for the smaller facial components (*e.g.*, eyes and lips) where with smaller resolution images, the segmentation network performs poorly. We note that the one-way connection-based SPN is quite successful on relatively simple tasks such as the HELEN dataset, but fails for more complex tasks, as revealed by the results of Pascal VOC dataset in the following section.

**Pascal VOC Dataset.** The PASCAL VOC 2012 segmentation benchmark [6] involves 20 foreground object classes and one background class. The original dataset contains 1464 training, 1499 validation and 1456 testing images, with pixel-level annotations. The performance is mainly measured in terms of pixel IoU averaged across the 21 classes. We train our SPNs on the train split with the coarse segmentation results produced by the FCN-8s model [21]. The model is fine-tuned on a pre-trained VGG-16 network, where different levels of features are upsampled and concatenated to obtain the final, low-resolution segmentation results ($8\times$ smaller than the original image size). The guidance network of the SPN also fine-tunes the VGG-16 structure from the beginning till the *pool5* layer as the downsampling part. Similar to the settings for the HELEN dataset, the upsampling part has a symmetric structure with skipped links until the feature dimensions of $64 \times 64$. The spatial propagation module has the same configuration as that of the SPN that we employed for the HELEN dataset. The model is applied on the coarse segmentation maps of the validation and test splits generated by any image segmentation algorithm without fine-tuning. We test the refinement SPN on three base models: (a) FCN-8s [21], (b) the atrous spatial pyramid pooling (ASPP-L) network fine-tuned with VGG-16, denoted as Deeplab VGG, and (c) the ASPP-L: a multi-scale network fine-tuned with ResNet-101 [11] (pre-trained on the COCO dataset), denoted as Deeplab ResNet-101. Among them, (b) and (c) are the two basic models from [5], which are then refined with dense CRF [14] conditioned on the original image.

Table 3 shows that through the three-way SPN, the accuarcy of segmentation is significantly improved over the coarse segmentation results for all the three baseline models. It has strong capability of generalization and can successfully refine any coarse maps from different pre-trained models by a large margin. Different with

Table 2: Quantitative comparison (mean IoU) with dense CRF-based refinement [5] on Deeplab pre-trained models.

| mIoU | CNN | +dense CRF | +SPN |
|---|---|---|---|
| VGG | 68.97 | 71.57 | **73.12** |
| ResNet | 76.40 | 77.69 | **79.76** |

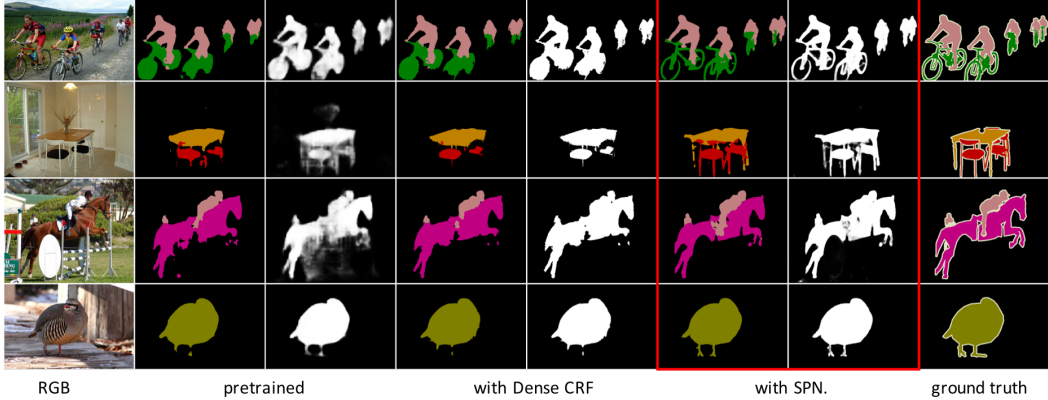

RGB        pretrained        with Dense CRF        with SPN.        ground truth

Figure 4: Visualization of Pascal VOC segmentation results (left) and object probability (by $1 - P_b$, $P_b$ is the probability of background). The "pretrained" column denotes the base Deeplab ResNet-101 model, while the rest 4 columns show the base model combined with the dense CRF [5] and the proposed SPN, respectively.

Table 3: Quantitative evaluation results on the Pascal VOC dataset. We compare the two connections of SPN with the corresponding pre-trained models, including: (a) FCN-8s (F), (b) Deeplab VGG (V) and (c) Deeplab ResNet-101 (R). AC denotes accuracy, "+" denote added on top of the base model.

| Model | F | +1 way | +3 way | V | +1 way | +3 way | R | +1 way | +3 way |
|---|---|---|---|---|---|---|---|---|---|
| overall AC | 91.22 | 90.64 | **92.90** | 92.61 | 92.16 | **93.83** | 94.63 | 94.12 | **95.49** |
| mean AC | 77.61 | 70.64 | **79.49** | 80.97 | 73.53 | **83.15** | 84.16 | 77.46 | **86.09** |
| **mean IoU** | 65.51 | 60.95 | **69.86** | 68.97 | 64.42 | **73.12** | 76.46 | 72.02 | **79.76** |

the Helen dataset, the one-way SPN fails to refine the segmentation, which is probably due to its limited capability of learning preferable affinity with a sparse form, especially when the data distribution gets more complex. Table 2 shows that by replacing the dense CRF module with the same refinement model, the performance is boosted by a large margin, without fine-tuning. One the test split, the DeepNet ResNet-101 based SPN achieves the mean IoU of **80.22**, while the dense CRF gets 79.7. The three-way SPN produces fine visual results, as shown in the red bounding box of Figure 4. By comparing the probability maps (column 3 versus 7), SPN exhibits fundamental improvement in object details, boundaries, and semantic integrity.

In addition, we show in table 4 that the same refinement model can also be generalize to dilated convolution based networks [34]. It significantly improves the quantitative performance on top of the "Front end" base model, as well as adding a multi-scale refinement module, denoted as "+Context". Specifically, the SPN improves the base model with much larger margin compared to the context aggregation module (see "+3 way" vs "+Context" in table 4).

## 6 Conclusion

We propose spatial propagation networks for learning pairwise affinities for vision tasks. It is a generic framework that can be applied to numerous tasks, and in this work we demonstrate its effectiveness for semantic object segmentation. Experiments on the HELEN face parsing and PASCAL VOC object semantic segmentation tasks show that the spatial propagation network is general, effective and efficient for generating high-quality segmentation results.

Table 4: Quantitative evaluation results on the Pascal VOC dataset. We refine the base models proposed with dilated convolutions [34]. "+" denotes additions on top of the "Front end" model.

| Model | Front end | +3 way | +Context | +Context+3 way |
|---|---|---|---|---|
| overall AC | 93.03 | 93.89 | 93.44 | **94.35** |
| mean AC | 80.31 | 83.47 | 80.97 | **83.98** |
| **mean IoU** | 69.75 | 73.14 | 71.86 | **75.28** |

**Acknowledgement.** This work is supported in part by the NSF CAREER Grant #1149783, gifts from Adobe and NVIDIA.

## Footnotes

[1]The original output (also for evaluation) size it $250 * 250$.

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
