[Supplementary Material]

# Supplementary Material: Learning Affinity via Spatial Propagation Networks

**Sifei Liu**
UC Merced, Nvidia

**Shalini De Mello**
Nvidia

**Jinwei Gu**
Nvidia

**Guangyu Zhong**
Dalian University of Technology

**Ming-Hsuan Yang**
UC Merced, Nvidia

**Jan Kautz**
Nvidia

## 1  More Experimental Settings

We combine the guidance network and the spatial propagation module similarly to [4]. We use two propagation units (*e.g.*, the bottom part in Figure 2 of the paper is one propagation unit) with cascaded connections to achieve better results. Differently, we feed in the integrated hidden map of the first unit to the second unit, instead of cascading each direction separately and integrate them at the end of the second unit. We use two more convolutional layers with 32 channels before and after the propagation units to transfer the input map to an intermediate feature map and to make it compatible with the node-wise max-pooling. In addition, we maintain a smaller size of the propagation layer to make the model more efficient w.r.t computational speed and memory. This is carried out by bi-linearly downsampling/upsampling after the two convolutional layers, so that the hidden maps of propagation module is with a smaller dimension of $64 \times 64$. Note that to compare the one-way with the three-way connection, we use exactly the same structure except the propagation units. We do not apply any configuration used by [1] or [4].

## 2  Proof of Theorem 3

**Theorem.** *Let* $\left\{ p_{t,k}^{K} \right\}_{k \in \mathbb{N}}$ *be the weights in* $w_t$. *The model can be stabilized if* $\sum_{k \in \mathbb{N}} \left| p_{t,k}^{K} \right| \leq 1$.

*Proof.* Let $\lambda$ be the eigenvalues of matrix $w_t$ and $\lambda_{max}$ be the largest one. According to Gershgorin's Theorem [2], when every eigenvalue of a square matrix $w_t$ satisfies:

$$|\lambda - p_{t,t}| \leq \sum_{k=1,k \neq t}^{n} |p_{k,t}|, \quad t \in [1, n] \tag{1}$$

then $|\lambda - p_{t,t}| + |p_{t,t}| \leq \sum_{k=1}^{n} |p_{k,t}|$. According to the triangular inequality, and since $\sum_{k=1,t \neq k}^{n} |p_{k,t}| \leq 1$, we have

$$\lambda_{max} \leq |\lambda - p_{t,t}| + |p_{t,t}| \leq \sum_{k=1}^{n} |p_{k,t}| \leq 1 \tag{2}$$

which satisfies the model stability condition. $\qquad\square$

Theorem 3 in the paper shows that the stability of a linear propagation model can be maintained by regularizing all the weights of each pixel in the hidden layer such the summation of their absolute values is less than one. For the one-way connection, Chen *et al.* [1] maintain each scalar output $p$ to be within $(0, 1)$. Liu *et al.* [4] extend the range to $(-1, 1)$, where the negative weights show preferable effects for learning image enhancers. This indicates that the affinity matrix is not necessarily restricted

Figure 1: Results of face parsing on the HELEN dataset with detailed regions cropped from the high resolution images. (Images are all in high resolution and can be viewed by zooming in.)

to be positive/semi-positive definite. (*e.g.*, this setting is also used for a pre-defined affinity matrix in [3].) For the three-way connection, we simply regularize the three weights (the output of a deep CNN) according to Theorem 3 without any positive/semi-positive definite restriction.

## 3 Additional Face Parsing results on the HELEN dataset

**Baseline network settings.** We first train a baseline CNN with a symmetric U-net structure, where both the input and output are $8\times$ smaller than the original image. The downsampling part of the network is equipped with five consecutive conv+relu+max-pooling (with stride of 2) layers. Starting from 32, each one has double the number of channels, resulting in a $4 \times 4 \times 512$ feature maps at the bottleneck. In order to use the information at different levels of image resolution, we add skipped-links by summing features maps of the same dimensions from the corresponding upsample and dowsample layers. The upsample part has symmetric configurations, except that the max-pooling is replaced with bilinear upsampling, and the last sub-module has 11 channels for the 11 classes.

**Qanlitative results.** We show more parsing results on the HELEN dataset. The detailed regions are cropped from the high resolution results. Figure 1 shows the effectiveness of the proposed spatial propagation network (SPN).

## 4 Semantic segmentation results on the PASCAL dataset

Specifically, the SPN improves the base model with much larger margin compared to the context aggregation module (see "+3 way" vs "+Context" in Table 4 of the paper).

Figure 2: Based on model R, we visualize the Pascal VOC segmentation results (left) and object probability (by $1 - P_b$, where $P_b$ denotes the probability of the background region).

**Qanlitative results.** We show more semantic segmentation results (left) and object probability (*i.e.*, $1 - P_b$, where $P_b$ denotes the probability of the background region) on the Pascal VOC 2012 dataset (Figure 2).