[Reviews · NeurIPS 2017]

Reviewer 1



The authors incorporate ideas from image processing into CNNs and show how nonlinear diffusion can be combined with deep learning. This allows to train more accurate post-processing modules for semantic segmentation, that are shown to outperform denseCRF-based post-processing, or recurrent alternatives that rely on more straightforward interpretations of recursive signal filtering, as introduced in [3,16]. The main practical contribution lies in extending the techniques of [3,16]: when these techniques apply recursive filtering, say in the horizontal direction of an image, they pass information along rows in isolation. Instead the method of the authors allows one to propagate information across rows, by rephrasing the originally scalar recursion in terms of vector-matrix products. This is shown to be much more effective than the baseline. On the one hand this is an interesting, and technically non-trivial development -it was therefore intriguing to read the paper. The results also seem positive. On the other hand the paper is written in a confusing manner. Even though I am quite familiar with recursive filtering and nonlinear diffusion I had to read the paper more than 4 times until I could understand what the notation is supposed to mean. Below is a sample of problems I have had with notation when reading this the first time (by now I do understand - but it is the authors responsibility to make the reading easy rather than have the reader guess): Eq. 1: w_t is used but not defined - is it a scalar or a matrix; d_t is used and defined in terms of the (undefined) w_t; Eq. 2: we get an expression for d_t(i,i) - but the range of i is not specified, and the values of d_t(i,j) i \neq j are not defined. Eq. 3: lambda_t appears (one more symbol), but is only defined after 5 lines. l. 102: we only now learn the dimensionality of w_t Theorem 1: even after reading 4 times I still do not know how this was useful (but I agree that it is correct). Even if I could understand it, I think it would be better to first show why this is needed, before moving on to prove it. And even better to put it in some appendix. Theorem 2: not directly clear what you mean by the "degree matrix composed of d_t"- it would be good to specify the dimensions of d_t and D, and how D is formed by d_1, d_N (I do understand, but I need to imagine) Results: even though there seem to be substantial improvements, I am not too sure about some things: - Table 3: if one-way connection is the same (or almost the same) as [3], how can it be that in your case you get worse results than the baseline, while they get better? -Table 1: the numbers are quite different from those reported in [3,4] -Table 2: It looks like even your baseline is better than [16] (due to a stronger network maybe?) But the most interesting thing is the added value that your method brings to the problem, not the exact number itself. It would be good if possible to run an experiment with the same starting point as [16]

Reviewer 2



The authors propose a Spatial Propagation Network to learn the affinity matrix with applications to semantic segmentation on HELEN and PASCAL VOC 2012 benchmarks. The proposed method with three way connections improves strong baselines on both datasets. Furthermore, the authors have provided detailed experimental results and analysis of the proposed model. The reviewer finds this work very interesting (very impressive visualization results on PASCAL VOC), and has only a few concerns: 1. Two propagation units are employed for experiments. Will the performance further improve with more units? 2. Is it possible to elaborate more on how Theorem 3 is enforced in the implementation, especially for the tree way connection case? 3. Is it possible to visualize the learned propagation strength, p (maybe for each connection way)? 4. Thanks for providing the inference time in the supplementary material. It may be interesting to also list the inference time consumed for the other components so that the readers have a better understanding of whole model inference time. 5. The proposed one-way SPN fails to refine the segmentation results on PASCAL VOC 2012. Is it possible to analyse the failure modes visually? Maybe it is because of the line-style artifacts, which can not even be removed with multiple propagation units? 6. Typos: line 173: Figure. 1(b) line 184: Figure 1(a) and 1(b) line 186: Figure 1(b) line 268: previous

Reviewer 3



The paper describes a method for learning pairwise affinities for recurrent label refinement in deep networks. A typical application is as follows: a feature map is produced by a convolutional network and is then refined by additional layers that in effect pass messages between pixels. The weights for such message passing are often set using hand-defined feature spaces (although prior work on learning such weights exists, see below). The submission describes a formulation for learning such weights. The paper has a number of issues that lead me to recommend rejection: 1. The general problem tackled in this paper -- refining poorly localized segmentation boundaries -- has been tackled in many publications. Two representative approaches are: (a) add layers that model mean field inference in a dense CRF and train them jointly with the initial segmentation network (as in [1,13,30]); (b) add a convolutional refinement module, such as the context module in [Multi-Scale Context Aggregation by Dilated Convolutions, ICLR 2016], also trained jointly with the segmentation network. The submission should provide controlled experiments that compare the presented approach to this prior work, but it doesn't. An attempt is made in Table 1, but it is deeply flawed. As far as I can tell, the dense CRF is not trained end-to-end with the segmentation network, as commonly done in the literature, such as [1,13,30]. And the context module (the ICLR 2016 work referred to above) is not compared to at all, even though it was developed for this specific purpose and is known to yield good results. (In fact, the ICLR 2016 paper reports refinement results on the VOC 2012 dataset with the VGG network that are better than the SPN results in Table 1 (IoU of 73.9 in the "Front + Large + RNN" condition in Table 3 of the ICLR 2016 paper). And that's a comparatively old work by now.) 2. There is other related work in the literature that specifically addresses learning affinities for label refinement. This work is closely related but is not cited, discussed, or compared to: -Semantic Segmentation with Boundary Neural Fields. Gedas Bertasius, Jianbo Shi and Lorenzo Torresani. CVPR 2016 - Convolutional Random Walk Networks for Semantic Image Segmentation. Gedas Bertasius, Lorenzo Torresani, Stella X. Yu, Jianbo Shi. CVPR 2017 - Learning Dense Convolutional Embeddings for Semantic Segmentation. Adam W. Harley, Konstantinos G. Derpanis, Iasonas Kokkinos. ICLR Workshop 2016 3. The results on VOC 2012 are well below the current state of the art, which stands at 86% IoU (compared to 79.8% in the submission). One could argue that the authors are more interested in evaluating the contribution of their refinement approach when added to some baseline segmentation networks, but (a) such controlled evaluation was not done properly (see point (1) above); and (b) the authors' combined approach is quite elaborate, so it would be hard to claim that it is somehow much simpler than state-of-the-art networks that dominate the leaderboard. With this level of complexity, it is reasonable to ask for state-of-the-art performance. Minor comment: - Since the dense CRF seems to play an important role in the submission, as a baseline that is repeatedly compared against, the submission should cite the dense CRF paper: [Efficient Inference in Fully Connected CRFs with Gaussian Edge Potentials, NIPS 2011].